# Virtual Screening Technology for Two Novel Peptides in Soybean as Inhibitors of α-Amylase and α-Glucosidase

**DOI:** 10.3390/foods12244387

**Published:** 2023-12-06

**Authors:** Xiyao Tang, Xu Chen, Hong Wang, Jinyi Yang, Lin Li, Jie Zhu, Yujia Liu

**Affiliations:** 1Key Laboratory of Healthy Food Development and Nutrition Regulation of China National Light Industry, School of Life and Health Technology, Dongguan University of Technology, Dongguan 523808, Chinayujialiu@dgut.edu.cn (Y.L.); 2College of Food Science, South China Agricultural University, Guangzhou 510642, China

**Keywords:** soybean peptides, α-amylase, α-glucosidase, molecular docking, inhibition activity

## Abstract

Soybean peptides (SPs) have bioactivities of enzyme inhibition that are beneficial to human health, but their mechanism is not clear. This study aimed to identify peptide fragments in SPs that simultaneously inhibit α-amylase and α-glucosidase and to explore their enzyme inhibition mechanism. Firstly, the inhibitory activity of SPs against the enzymes was determined. And two octapeptides, LDQTPRVF and SRNPIYSN, were identified for the first time by using HPLC-QTOF-MS/MS and virtual screening. Molecular simulation results showed that hydrogen bonds and π–π bonds were the key factors, and the N-terminal (Leu and Ser) and C-terminal (Phe) of peptide were important inhibiting sites. Both octapeptides were synthesized, and their IC50 values were 3.08 and 5.58 mmol/L for α-amylase, and 2.52 and 4.57 mmol/L for α-glucosidase, respectively. This study provided evidence for SPs as a potential inhibitor of α-amylase and α-glucosidase in special dietary foods.

## 1. Introduction

Diabetes mellitus (DM) is a common chronic metabolic disease that is generally associated with serious complications, such as retinopathy, nephropathy, neuropathy, cerebrovascular disease, heart disease, and lower-extremity arterial disease. The global incidence of DM is increasing rapidly, and it is estimated that 783 million people will suffer from DM by 2045 [1]. In 1999, the World Health Organization classified DM into four types: type I diabetes mellitus (T1DM), type II diabetes mellitus (T2DM), special-type diabetes mellitus, and gestational diabetes mellitus (GDM). Globally, T2DM is the most prevalent type of diabetes. The prevalence of T2DM is closely related to social development, population aging, obesity rate, and genetic predisposition of the population [2]. The pathophysiology of T2DM is characterized by a decrease in insulin-regulated glucose metabolism (i.e., insulin resistance) accompanied by a decrease (or relative decrease) in insulin secretion due to defective islet b-cell function [3].

The significant increase in postprandial blood glucose is unfavorable for patients with diabetes. Research has shown that reducing the intake of rapidly digested starch and delaying the digestion time of carbohydrates is effective in alleviating the phenomenon of elevated blood sugar in the body [4]. α-Amylase and α-glucosidase are key enzymes in the digestion of dietary starch in the human body, and they can ultimately convert starch into glucose in the digestive tract [5]. α-Amylase hydrolyzes starch into dextrin, oligosaccharides, and glucose. α-glucosidase is one of the key enzymes that decompose disaccharides into glucose [6]. Inhibition of these enzymes can delay the hydrolysis of dietary starch in the digestive system, lower blood glucose levels, slow down glucose metabolism, and delay glucose absorption [7]. At present, acarbose is used as a glucosidase inhibitor, but long-term use of this drug can cause gastrointestinal side effects. Therefore, the development of natural sources of hypoglycemic enzyme inhibitors with fewer side effects is important.

Peptides are obtained from the decomposition of proteins. Standard methods for preparing peptides in the food industry include enzymatic hydrolysis and microbial hydrolysis [8]. Based on the differences in some parameters involving structure, charge, amino acid sequences, and interactions, peptides exhibit some special biological activities [9,10]. It is reported that antidiabetic peptides play a role through one or more mechanisms, including inhibition of enzymes (i.e., digestive enzymes linked to type 2 diabetes and DPP-IV), regulation and improvement of insulin secretion and the effect of auxin, as well as control of satiety and gastric emptying [6]. On the aspect of inhibiting glucose-digesting enzymes (mainly amylase and glucosidase) related to type 2 diabetes, peptides from various sources or peptides from the same source with different structures may have different inhibitory mechanisms on enzymes. The inhibitory mechanism of peptides on amylase and glucosidase has been studied from the perspective of inhibition kinetics, molecular simulation, and spectrum [11,12]. Studies have shown that peptides could inhibit enzyme activity through competitive inhibition. For example, a novel peptide, GPPGPA, identified from the skin collagen of the Chinese giant salamander, has competitive inhibitory effects on α-glucosidase [13], which means that it is able to compete with the substrate for the same binding site in the reaction. However, peptides from the same protein source obtained under different conditions also have similar enzyme inhibition mechanisms. For instance, the quinoa protein [14], digested and hydrolyzed in vitro, showed competitive inhibition effects on α-glucosidase regardless of whether the raw material was from germinated or non-germinated quinoa. Moreover, noncompetitive inhibition or mixed inhibition also exists. Two oligopeptides, GGSK and ELS, isolated from *Porphyra* spp., could still bind with the substrate after binding with α-amylase [11]; LLLLPSYSEF identified from Camellia seed cake presents mixed inhibition on α-glucosidase, with IC50 of 1.11 mmol/L [15]. In terms of intermolecular interactions, amino acid residues in the enzyme active cavity are prone to form hydrogen bonds with amino acids on peptides, and more and shorter hydrogen bonds indicate a more stable interaction between the two [16]. Three peptides, LDLQR, AGGFR, and LDNFR, which were identified from wheat peptides, had strong hydrogen bonds with residues Asp 203, Gln 603, and so on in the active site of α-glucosidase [17]. Under special binding configurations, the π–π bond was easily formed between aromatic amino acid residues of peptides and enzymes [18]. Furthermore, van der Waals forces, electrostatic forces, and hydrophobic interactions were key interactions between peptides and enzymes [12]. In recent years, most studies have focused on the inhibitory activity of bioactive peptides on a single glucose-digesting enzyme, but there have also been studies indicating that peptides hydrolyzed from the same protein source have certain inhibitory effects on both amylase and glucosidase. The peptide segment KLPGF identified in egg albumin had good inhibitory activity against α-amylase and α-glucosidase, with IC50 values of 120.0 and 59.5 μmol/L, respectively [19]. Since α-amylase and α-glucosidase existed as key metabolic enzymes in the process of carbohydrate digestion, it was significant to find bioactive peptides that had good inhibitory activity against both glucose-digesting enzymes from the perspective of high efficiency in the prevention and treatment of diabetes.

Soybean has high nutritional value and is an important source of plant-derived protein in daily diet. At present, many studies have proven that soybean peptides have many physiological activities, such as anti-hypertensive, antibacterial, and anti-oxidant activities [20,21]. From an anti-diabetes point of view, the main way by which soybean peptides affect starch digestion is by changing the properties of starch substrate [22] and inhibiting the activity of starch digestive enzymes [23]. Previously, the inhibitory effect of soybean crude peptides on amylase has been verified through fluorescence and inhibition of kinetic methods [12]. On this basis, the aim of this study was to explore which bioactive peptide segments in soybean crude peptides have inhibitory activity against both α-amylase and α-glucosidase, as well as to elucidate their inhibitory effects and molecular mechanisms. First, the inhibitory effect of soybean crude peptides on two enzymes was measured, and then the peptide segments in soybean crude peptides were identified by high-performance liquid chromatography–secondary mass spectrometry (HPLC–MS/MS). The obtained peptide fragments were then virtually screened. Two soybean oligopeptides were obtained according to the prediction results, and their enzyme inhibition was verified. Finally, molecular docking was used to clarify their inhibition mechanism on enzymes. This study provides some theoretical support for the application of soybean peptides as potential inhibitors of α-amylase and α-glucosidase in special dietary foods.

## 2. Materials and Methods

### 2.1. Chemicals

p-Nitrophenyl-α-D-glucopyranoside (pNPG) and soy protein isolate were obtained from Mackiln (Shanghai, China). DL-dithiothreitol (DTT), iodoacetamide (IAA), formic acid (FA), acetonitrile (ACN), methanol, and porcine pancreatic amylase (PPA) (>10 units/mg) were acquired from Sigma-Aldrich (St. Louis, MO, USA). α-Glucosidase (from Saccharomyces cerevisiae, 26.5 units/mg) and trypsin (from porcine pancreas, 250 units/mg) were obtained from Yuanye Biotech (Shanghai, China). DNS solution was purchased from Rhawn (Shanghai, China). Oligopeptides were prepared with LifeTein (Beijing, China). High-performance liquid chromatography (HPLC)-grade acetonitrile, methanol, and formic acid were obtained from Merck (Darmstadt, Germany). The other chemicals were of analytical grade.

### 2.2. Methods

Soybean peptide solution was prepared using the following methods [24]: soy protein solution was subjected to denaturation by treatment at 100 °C for 10 min, followed by enzymatic hydrolysis using trypsin under reaction conditions of pH 7 and 37 °C. The hydrolysis products were then heat-inactivated in a boiling water bath, and the precipitate was subsequently removed by centrifugation. The supernatant was collected and freeze-dried to obtain crude soy peptides for further use.

#### 2.2.1. α-Amylase Inhibition Assay In Vitro

Methods refer to Admassu’s and make appropriate modifications [11]. The starch solution (10 mg/mL), enzyme solution (0.5 U/mL), and soybean peptide solution at different concentrations (5, 10, 15, 20, 25 mg/mL) were prepared, the solvent was PBS (pH 6.9–7.2). The starch solution was gelatinized for 30 min in a water bath at 100 °C and then kept at 37 °C. Enzyme solution and peptide solution of different concentrations were premixed and preheated at 37 °C for 20 min (Sample group). The peptide solution was replaced by equal volume PBS solution in the blank group (The operations in Section 2.2.2 are the same). At the beginning of the experiment, the starch solution was added to the mixed liquor. After incubation at 37 °C for 10 min, 200 μL of the reaction solution was added to 400 μL of the DNS solution. All samples were placed in a 100 °C bath for 5 min and then held at indoor temperature. After addition of 1 mL of deionized water into each sample and vortexing, 200 μL of the sample solution was added into 96-well plates, with three parallels in each group. The absorbance was measured at 540 nm, and the α-amylase inhibition ratio was calculated according to Formula (1):(1)$$\alpha -Amylase\;Inhibition\;Ratio=\left(1-\frac{A_{1}-A_{2}}{A_{3}-A_{4}}\right)\times 100\%$$
where A_1_, A_2_, A_3_, and A_4_ are the absorbance values of the sample, sample blank, negative control, and blank, respectively.

#### 2.2.2. α-Glucosidase Inhibition Assay In Vitro

Methods refer to Zhang’s, with appropriate modifications [15]. pNPG solution (1 mmol/L), enzyme solution (0.25 U/mL), and soybean peptide solution at different concentrations (5, 10, 15, 20, 25 mg/mL) were prepared, and the solvent was PBS (pH 6.9–7.2). All reagents were preheated at 37 °C for 20 min. At the beginning of the experiment, pNPG solution was added to all experimental groups. After incubation at 37 °C for 10 min, 200 μL of reaction solution was added to 1 mL of deionized water and boiling water bath for 1 min to stop the reaction. Subsequently, 200 μL of sample solution was transferred into a 96-well plate, with three parallel plates in each group. The absorbance at 400 nm was measured, and the α-glucosidase inhibition ratio was calculated according to Formula (2):(2)$$\alpha -Glucosidase\;Inhibition\;Ratio=\left(1-\frac{A_{1}-A_{2}}{A_{3}-A_{4}}\right)\times 100\%$$
where A_1_, A_2_, A_3_, and A_4_ are the absorbance values of the sample, sample blank, negative control, and blank, respectively.

#### 2.2.3. Structural Identification of Soybean Peptides (SPs)

The structural identification of SPs has referred to Liu’s method [17]. Peptides were reduced at 56 °C with 10 mmol/L DTT for 1 h and alkylated with 50 mmol/L IAA for 40 min at room temperature in the dark. The peptides were then lyophilized and resuspended in 20 μL of 0.1% formic acid. After that, HPLC–MS/MS analysis was performed using a 2.1 mm × 100 mm HSS T3 column packed with a reversed-phase C18 resin (1.8 μm, 100 Å, Acquity UPLC, Waters, Dublin, Ireland). Chromatographic separation was performed as follows. Mobile phase A (0.1% trifluoroacetic in 100% water, *v*/*v*) and elution B (0.1% trifluoroacetic in 100% acetonitrile, *v*/*v*) were used in a linear gradient procedure: 4–8% B from 0 to 2 min, 8–28% B from 2 to 45 min, 28–40% B from 45 to 55 min, 40–95% B from 55 to 56 min, and elution B at 95% for 10 min. Mass spectrometry (MS) analysis was performed using a four-stage rod time-of-flight (QTOF) tandem mass spectrometer with collision-induced dissociation (CID) as the activation type. The operating parameters were set as follows: positive polarity, 5.5 kV spray voltage, 0.25 mL/min flow rate, and 25 V collision energy for all peaks. Spectra were recorded over a mass/charge (Da) range of 50–1000.

#### 2.2.4. Virtual Screening

Virtual screening, also known as computer screening, uses molecular docking software on a computer to simulate the interaction between the target and the polypeptide before screening, as well as calculate the affinity of reducing the actual number of screened compounds. AutoDock Vina v1.1.2 software was used to screen for potential peptides. The structures of receptors were downloaded using RCSB PDB (https://www.rcsb.org accessed on 5 Jun 2023). α-Amylase (PDB ID:5U3A, resolution: 0.95 Å) and α-Glucosidase (PDB ID: 2QLY, resolution: 2.00 Å) were selected as the target protein crystal structures because of their *Homo sapiens* attribute. In addition, 5U3A and 2QLY have been used in α-amylase and α-glucosidase inhibitor studies elsewhere, respectively [25,26]. Proteins were treated with Schrodinger’s Preparation Wizard. First, the crystal water was removed, the missing hydrogen atom was added, and then the energy was minimized to optimize the geometry. All the above processes were run on AutoDockTools 1.5.7. Acarbose was used as positive control in the screening process. PSMS > 2 and IonScore > 40 were used as screening criteria.

#### 2.2.5. Molecular Docking

In this study, molecular docking simulation was used to predict the interaction and binding patterns of different soybean peptides with α-amylase and α-glucosidase. The protein structures of two enzymes were obtained from RCSB PDB. The three-dimensional conformation of soybean peptides was constructed and optimized by PyMOL 2.2.0 software. Prior to docking, the ligand (soy peptides) and the receptor (α-amylase, α-glucosidase) were processed by using AutoDockTools 1.5.7, the presence of small molecules and non-polar water molecules was removed from the crystal structure of the protein, and hydrogen atoms and Gasteiger–Huckel charges were added. The docking pocket was arranged to completely wrap the active cavity of the enzyme. We used AutoDock program to study the molecular binding of inhibitory ligands at amylase binding sites, and we performed 100 assays using genetic algorithms [27]. The Lamarckian GA module was used to sort all output results according to the size of the docking energy, and the other options were maintained at default parameters.

#### 2.2.6. Prediction of Physicochemical Properties

Using established information databases to predict the molecular weight, isoelectric point, toxicity, protein source, and potential biological activities of bioactive peptides is one of the widely applied methods in the field of bioinformatics. This computer prediction method has both efficiency and accuracy. The physicochemical properties of novel peptides were predicted by Expay (https://web.expasy.org/compute_pi/, accessed on 5 Jun 2023), Innovagen (http://www.innovagen.com/proteomics-tools, accessed on 6 Jun 2023) and Pepdraw (http://pepdraw.com/, accessed on 5 Jun 2023). The protein source was searched by Uniprot (https://www.uniprot.org/blast, accessed on 6 Jun 2023), and the toxicity of bioactive peptides was forecasted using ToxinPred (https://webs.iiitd.edu.in/raghava/toxinpred/protein.php, accessed on 6 Jun 2023).

#### 2.2.7. Statistical Analysis

The results in this study were plotted using OriginPro 9.8.0, and the results are expressed as the mean ± standard error of the mean. The significance of comparisons between samples was analyzed by one-way ANOVA with a significance level of *p* < 0.05. Molecular docking images were completed by PyMOL 2.2.0 application and Chimera X v1.6.1 software.

## 3. Results

### 3.1. Inhibitory Effects of Crude SPs for α-Amylase and α-Glucosidase In Vitro

Crude peptides were hydrolyzed from soybean protein based on our previous studies [28]. Many studies have shown that peptides can inhibit glucose-digesting digestion enzymes [23,29]. As shown in Figure 1, soybean crude peptides had an inhibitory effect on porcine pancreatic α-amylase and α-glucosidase, and the inhibitory activity was proportional to the concentration of the SPs, with semi-inhibitory concentrations of 18.795 ± 0.118 mg/mL and 11.033 ± 0.033 mg/mL, respectively. The results indicated that soybean crude peptides had a certain inhibitory effect on both enzymes, which was similar to existing research results [12]. However, soybean crude peptides contain many unknown peptide segments because of differences in amino acid composition, amino acid quantity, and spatial conformation between each peptide segment, and there may be significant or minor differences in activity [28]. Some peptide segments may only have good inhibitory activity against one type of glucose-digesting enzyme, while others may have no inhibitory effect on glucose-digesting enzymes. Therefore, further research is needed to discover secure specific peptide segments that can simultaneously inhibit α-amylase and α-glucosidase.

### 3.2. Peptide Fragment Screening with α-Amylase and α-Glucosidase Inhibitory Activity

Because of its high specificity and accuracy, HPLC–MS/MS has been widely used in the isolation and identification of drug and food components [30,31]. By scanning the MS of soybean peptides and collecting the data continuously, a mass spectrogram was obtained in each scan, and then the longitudinal coordinates (ion strength) of each mass spectrogram were added to obtain the total ion strength. Finally, the total ion chromatogram (Appendix A) was constructed on the longitudinal and transverse coordinates of the total ion strength.

To identify peptides with α-amylase and α-glucosidase inhibitory activity, the peptide sequence of soybean crude peptides was identified by HPLC–QTOF-MS/MS, and 9246 initial peptides were recognized. PSMS represents the number of peptide segments that matched the secondary mass spectra; the higher the PSMS value, the more times the peptide segment was identified. Ionscore is the score of peptides matching the map, and an identity score can be obtained according to the number of candidate peptide segments and their Ionscore. When the Ionscore is lower than the identity score, the peptide segment can be evaluated as an untrusted peptide segment, and it would no longer be used for subsequent virtual screening work. Consequently, 202 peptides were obtained with PSMS ≥ 2 and Ionscore ≥ 40 as screening conditions, which were used as virtual screening [32]. Finally, the 202 peptides were compared with the Byonic database, and the binding energy of peptides to target receptors was used as the main screening condition to select the bioactive peptide that we needed.

Detailed information on 202 peptides is shown in Appendix A. The AutoDock Vina software was used to dock each peptide segment with α-amylase and α-glucosidase, and the corresponding binding energy was obtained. Subsequently, 202 peptides were sorted from small to large in terms of binding energy for docking with α-amylase and α-glucosidase, and peptides with the top 10% of the binding energy were selected for analysis through visual screening. The affinity between the ligand and receptor increases as the binding energy decreases. The lower binding energy means a better binding capacity. Finally, it was found that only two peptides, LDQTPRVF and SRNPIYSN, existed simultaneously in the first 10% binding-energy region of the two enzymes (Figure 2).

We obtained two octapeptides, LDQTPRVF (Figure 3a) and SRNPIYSN (Figure 3b), which had the inhibition effect of two enzymes and low binding energy. The energies for binding to α-amylase and α-glucosidase were −9.1 and −8.7 kcal/mol for LDQTPRVF and −7.6 and −7.5 kcal/mol for SRNPIYSN, respectively. Bioactive peptides can inhibit enzymatic activity interactions due to their small size and specificity [33]. The physicochemical properties of the two novel peptides were predicted, as shown in Table 1. The molecular weight, isoelectric point, hydrophobicity, and toxicity of the two peptides were predicted according to the amino acid species, number, and amino acid sequence of peptides. GRAVY is positive for hydrophobic, negative for hydrophilic, and between 0.5 and −0.5 for mainly amphoteric amino acids. On this basis, LDQTPRVF is an amphiphilic peptide, and SRNPIYSN has good hydrophilicity, which was verified in the actual experimental process. Furthermore, toxicity prediction showed that none of the two peptides were toxic. In addition, it was predicted that SRNPIYSN has an extinction coefficient probably because it contains tyrosine, which has strong ultraviolet absorption [34]. Moreover, by comparing the two peptides with the information in the known protein database using the Basic Local Alignment Search Tool (BLAST), it was confirmed they were both from Glycine max (soybean).

### 3.3. Molecular Docking for Screened Peptides

Molecular docking can identify the exact binding site of enzymes and peptides, and it can calculate the length of hydrogen bonds to help judge the interaction between a ligand and a receptor. In theory, inhibitory peptides can inactivate the enzyme by occupying the active site and blocking access to the active pocket. α-Amylase mainly consists of three structural domains, A, B, and C, with domain A showing a (β/α)8 TIM barrel structure, including the active site residues Asp197, Glu233, and Asp300. The barrel structure is made up of eight β-sheets surrounded by eight α-helixes; it has a certain catalytic ability, and it exists in all types of α-amylases. The structural domains B and C are highly symmetric in space about the structural domain A, and the structural domain B has three β-sheet and three α-helix, the length and structure of which vary depending on the source. Domain C mainly consists of antiparallel β structural composition due to its distance from the active center, has fewer mutation-prone sites, and can protect the stability of hydrophobic amino acids in the catalytic center [35,36]. To better distinguish the binding site of α-glucosidase, its amino acid sequence is divided into several regions, such as trefoil Type-P domain, N-terminal domain, catalytic (β/α)8 domain, catalytic domain Insert 1, catalytic domain Insert 2, proximal C-terminal domain (ProxC), and distal C-terminal domain (DistC) [37]. The active pocket of the enzyme is mainly formed by C-terminal β-strand residues such as Asp 203, Asp 443, and Asp 542, which are in the (β/α)8 barrel structure [17,20]. Binding to the active site of the enzyme or more hydrogen binding to the active region of the enzyme also increased the rate of inhibition of the enzyme by peptides. The docking results of the two peptides bound to α-amylase and α-glucosidase are shown in Figure 4 and Figure 5.

For α-amylase, the novel peptides form hydrogen bonds with residues such as Thr 163, His 201, His 305, and Asp 300, similar to previous research results [35]. The average binding distance between the two peptides and receptor are 2.09 and 2.07 Å, respectively, which are shorter than the traditional hydrogen bonding distance of 3.5 Å (Figure 4a–d). But, LDQTPRVF has more docking residues than SRNPIYSN, including Lys 200, Glu 233, and Asp 356. In the docking process, we found that both peptides bind in the active pocket of α-amylase, indicating that they inhibit substrate hydrolysis mainly by competing with the substrate for binding sites. Furthermore, phenylalanine residues on LDQTPRVF have a π–π interaction with residue Tyr 151. More receptor residues and more molecular interactions may be the reasons why LDQTPRVF has a better inhibitory effect on α-amylase than SRNPIYSN [38].

A total of seven hydrogen bonds were formed between LDQTPRVF and α-glucosidase with six residues (Glu 114, His 115, Trp 490, Lys 513, Lys 534, and Asp 777). The residue Asp 777 was connected to the peptide by two hydrogen bonds with bond lengths of both 1.9 Å, and the average length of all hydrogen bonds was 2.06 Å. A total of eight hydrogen bonds were formed between SRNPIYSN and α-glucosidase with residues Arg 202, Asp 203, Thr 205, Asn 207, Gln 603, and Tyr 605, similar to the results of existing studies [17], and the average length of all hydrogen bonds was 2.21 Å (Figure 5a–d). It is worth mentioning that the binding positions of the two peptides with α-glucosidase were different, LDQTPRVF docking at the inactive pocket area of the enzyme, indicating that its enzyme inhibition type was noncompetitive inhibition, uncompetitive inhibition or mixed inhibition, and SRNPIYSN binding at the entrance of the enzyme active pocket, whose presence could block the substrate from entering the active cavity of the enzyme. In addition, research has found that Leu and Ser at the N-terminus and Phe at the C-terminus of the peptides may be an important feature of peptide segments with carbohydrate-digestion enzyme-inhibitory activity [17]. This is because the amino acid residues at the ends of the peptide have more flexible spatial properties and can bind more tightly to the enzyme. Van der Waals forces and hydrophobic interactions are also interactions between the two novel peptides and two enzymes.

### 3.4. Inhibitory Activity Verification for Screened Peptides In Vitro

To verify the accuracy of the virtual screening results, we determined the IC50 values of α-amylase and α-glucosidase inhibitory activity of these synthetic peptides. As shown in Figure 6, the IC50 values of LDQTPRVF and SRNPIYSN for α-amylase are 3.08 ± 0.15 and 5.58 ± 0.13 mmol/L and, for α-glucosidase, they are 2.52 ± 0.04 and 4.57 ± 0.10 mmol/L, respectively. All results are concentration-dependent. There are reports that the IC50 of peptides identified from wheat germ for glucosidase is 8.59 mmol/L [17], and the IC50 of the peptide isolated and identified from enzymatic protein hydrolysates of red seaweed for amylase is 2.58 mmol/L [11]. These data are similar to the results of this study. According to reports, acarbose, an AGI hypoglycemic drug for clinical use, has IC50 of 0.45 mmol/L for α-amylase [11] and 60.8 μM for α-glucosidase [19]. In this regard, although the hypoglycemic activity of the two original bioactive peptides identified by our study could not achieve a drug effect, the research results revealed that soy peptides may be a potential, relatively safe, and functional food that can be consumed for a long time to assist in the treatment of diabetes. Furthermore, the results of IC50 for the two octapeptides can be consistent with the combination of virtual screening, indicating that the virtual screening method is reliable and can filter potential α-amylase and α-glucosidase inhibitors.

## 4. Discussion

At present, virtual screening is widely used in the fields of drug discovery and bioactive peptide research, providing a fast, efficient, and economical method for finding active substances with high affinity and selectivity. In addition, through simulation and model calculations, virtual screening technology can also predict the binding mode, stability, and activity of peptides with target proteins. This approach could help researchers gain insight into the mechanisms of action of bioactive peptides and further optimize and improve their function. Although bioactive peptides do not have the same enzyme inhibitory effect compared with drugs, they have a higher safety, and they can serve as auxiliary functional foods to alleviate chronic diseases. Currently, the research on the two novel glucose digestion enzyme inhibitory peptides mentioned in this article is still limited to the peptide–enzyme interaction system and the in vitro experimental stage. Subsequently, a deeper investigation will be conducted on the interaction phenomena between peptides and substrates, as well as the entire enzyme inhibition system, and attempts will be made to validate the activity of the peptides from an in vivo experimental perspective. In brief, the development of bioactive peptides is significant.

## 5. Conclusions

In this study, two soybean-derived bioactive peptides that have potential antidiabetic effects, LDQTPRVF and SRNPIYSN, were obtained by virtual screening. The results of enzyme inhibition experiments showed that the two novel peptides had inhibitory effects on α-amylase and α-glucosidase, which was also consistent with the results of virtual screening. The results of molecular docking showed the possible docking sites between the two peptides and two enzymes, and the docking positions showed the approximate type of inhibition of the peptides on the enzymes. Hydrogen bond is the main interaction force between peptides and enzymes, and there are hydrophobic interactions, van der Waals forces, and π–π interactions that contribute to the interaction between them. Leu and Ser at the N-terminus and Phe at the C-terminus of the peptides may be important features of peptide segments with starch-digestion enzyme-inhibitory activity. To conclude, soybean peptides can be considered as a candidate functional food for inhibiting the α-amylase and α-glucosidase, which is of significance for improving the utilization value of soybean in the diet. The bioactive peptides in food are valuable functional agents in healthy diets, but currently, most of the identified bioactive peptides are still in the in vitro research stage, and there is still a long way to go to widely apply bioactive peptides in daily diets.

## Figures and Tables

**Figure 1 foods-12-04387-f001:**
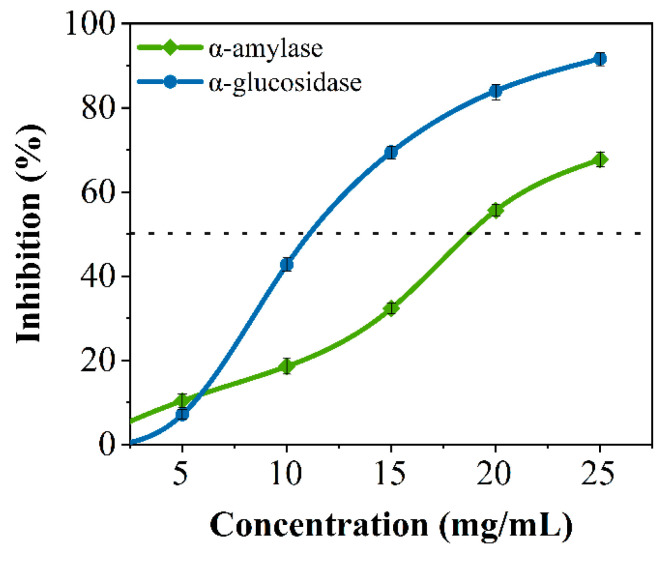
The inhibitory effects of crude soybean peptides on α-amylase (Green line) and α-glucosidase (Blue line) .

**Figure 2 foods-12-04387-f002:**
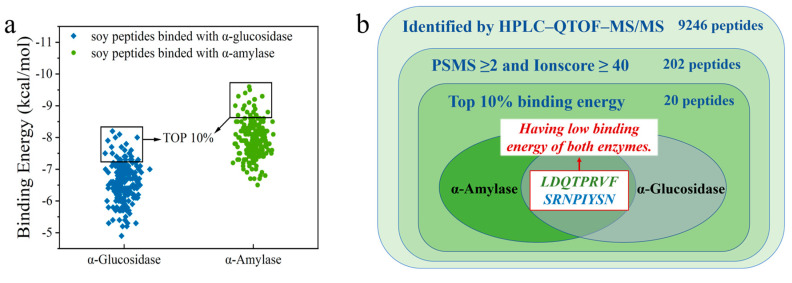
(**a**) Binding energy of peptide with α-amylase and α-glucosidase; (**b**) Venn diagram for peptide screening.

**Figure 3 foods-12-04387-f003:**
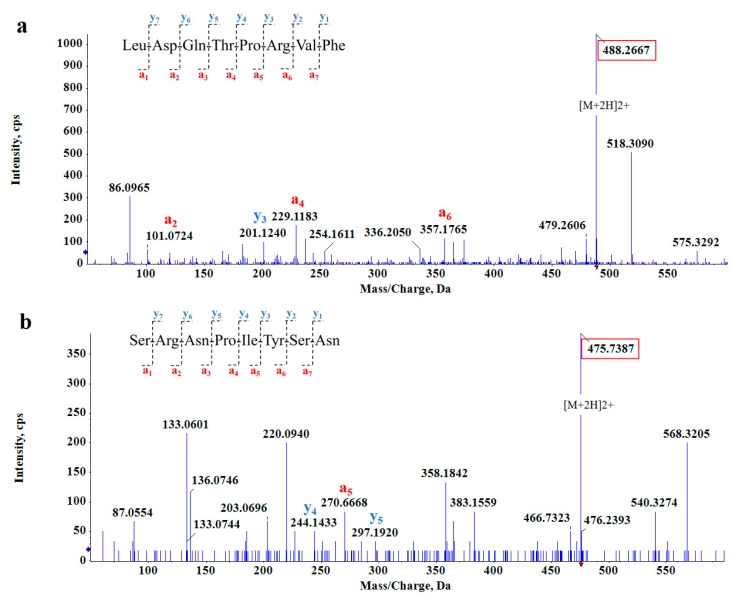
MS/MS spectrum of (**a**) LDQTPRVF and (**b**) SRNPIYSN.

**Figure 4 foods-12-04387-f004:**
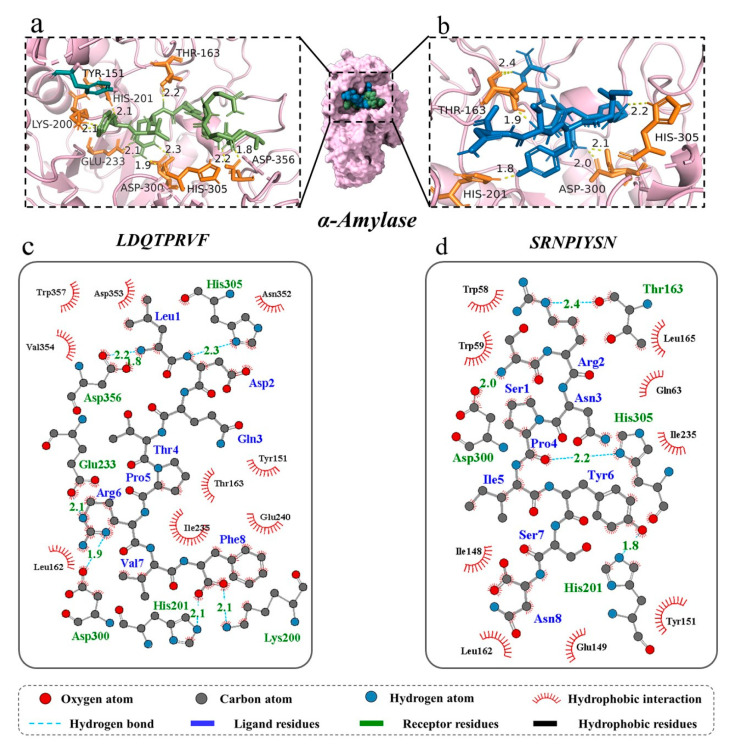
Three-dimensional and two-dimensional molecular docking diagram of the peptides of (**a**,**c**) LDQTPRVF and (**b**,**d**) SRNPIYSN with α-amylase.

**Figure 5 foods-12-04387-f005:**
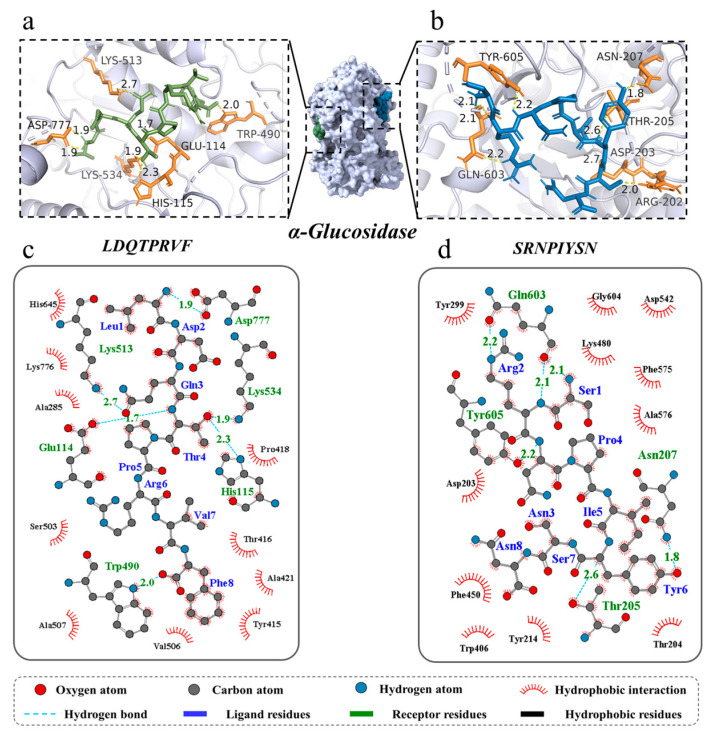
Three-dimensional and two-dimensional molecular docking diagram of the peptides of (**a**,**c**) LDQTPRVF and (**b**,**d**) SRNPIYSN with α-glucosidase.

**Figure 6 foods-12-04387-f006:**
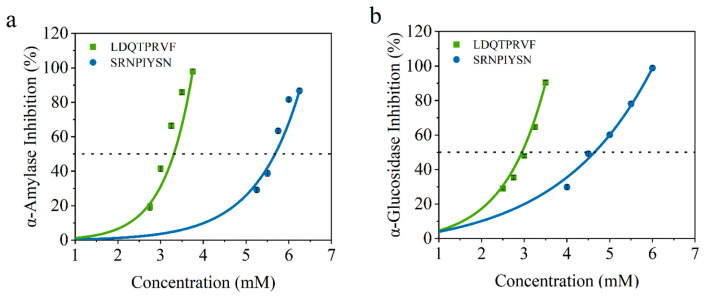
Inhibition activity of LDQTPRVF (Green line) and SRNPIYSN (Blue line) against (**a**) α-amylase and (**b**) α-glucosidase.

**Table 1 foods-12-04387-t001:** Physicochemical properties of the two peptides predicted by software.

PEPTIDES	LDQTPRVF	SRNPIYSN
Calculated mass (Da)	975.10	950.01
Observed mass (Da)	974.90	949.80
Isoelectric point	pH 6.64	pH 9.57
Grand average of hydropathicity (GRAVY)	−0.375	−1.438
Toxicity	Non-toxic	Non-toxic
Extinction coefficient (M^−1^ cm^−1^)	0	1280

## Data Availability

The data used to support the findings of this study can be made available upon request from the corresponding author. The data are not publicly available due to information that could compromise research participant privacy.

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
