# Peer review of "Virtual Screening Technology for Two Novel Peptides in Soybean as Inhibitors of α-Amylase and α-Glucosidase"

_foods, 2023, doi:10.3390/foods12244387_

Round 1

Reviewer 1 Report

Comments and Suggestions for Authors

foods-2731501

Title: Virtual screening technology for two novel peptides in soybean as inhibitors of α-amylase and α-glucosidase

Specific comments

1.     Line 13: “firstly” should not be placed in the end of the phrase.

2.     Lines 31-35: where is the reference for these statements?

3.     Line 52: “small molecular properties” is not clear, please rewrite. Peptides do not have several activities and functions because they are small. It involves many parameters in the structure, charge, interactions, etc.

4.     Line 70: Porphyra is the species, and it must be written in italics.

5.     Lines 110-124 contain comments from the MDPI’s template. Article should have been revised and fully read before evaluation by reviewers.

6.     Line 136 and 149: Where does these methods come from? Cite reference.

7.     Line 165: Why no information about the column is provided?

8.     Line 181: Homo sapiens is also a species and should be in italics. Please verify this in the entire paper.

9.     Lines 212-215: No comparison of means was performed? Please clarify which tests were used to compare the data.

10.   Line 230: Where does the toxicity potential come from?

11.   Figure 1: Why did the authors not test higher concentrations? An inversion may be happening after the highest concentration, or not.

12.   Line 264: Explanation to why low-binding energy peptides were chosen has to be added.

13.   Line 372: Where does this comment come from? Which data supports it?

14.   The fact that prediction suggested that peptides do not present toxicity is interesting, but still toxicity tests are the ultimate check. Why did the authors not check it?

15.   Authors defend in the discussion that peptides from soy could be used inside their food systems against diabetes but the IC50 was not tested in the original food matrix. So, in future work the effect of these peptides should be checked in the original matrix.

16.   Line 383: First phrase is out of context and does not make a lot of sense. What does nutrition have to do with the paper?

Comments on the Quality of English Language

Please check comments above.

Author Response

RESPONDS TO REVIEWER #1 COMMENTS:

1.Line 13: firstly should not be placed in the end of the phrase

Response: Thank you for the reviewer's comment. The corresponding content in the article has been revised. (Line 13)

2.Lines 31-35: where is the reference for these statements?

Response: Thanks for the reviewer's comment. We have supplemented some according references in the revised manuscript. (Line 31-35)

(1) Zheng, Y.; Ley, S. H. and Hu, F. B. Global aetiology and epidemiology of type 2 diabetes mellitus and its complications. Nat. Rev. Endocrinol. 2018, 14(2), 88-98

(2) Lin, Y. and Sun, Z. Current views on type 2 diabetes. J. Endocrinol. 2010, 204(1), 1-11

3.Line 52: small molecular properties is not clear, please rewrite. Peptides do not have several activities and functions because they are small. It involves many parameters in the structure, charge, interactions, etc.

Response: Thank you for the reviewer's suggestion. The corresponding content in the article has been revised. (Line 51-52)

4.Line 70: Porphyra is the species, and it must be written in italics.

Response: Thanks for the reviewer's comment. The corresponding content in the article has been revised. (Line 70, 436)

5.Lines 110-124 contain comments from the MDPIs template. Article should have been revised and fully read before evaluation by reviewers.

Response: Thanks for the reviewer’s suggest. The corresponding content in the article has been deleted. (Line 109)

6.Line 136 and 149: Where does these methods come from? Cite reference.

Response: These methods were referenced from:

(1) Admassu, H.; et al. Identification of Bioactive Peptides with alpha-Amylase Inhibitory Potential from Enzymatic Protein Hydrolysates of Red Seaweed (Porphyra spp). J. Agric. Food Chem. 2018. 66(19), 4872-4882. (Line 128)

(2) Zhang, Y.P.; et al. Optimization and Molecular Mechanism of Novel α-Glucosidase Inhibitory Peptides Derived from Camellia Seed Cake through Enzymatic Hydrolysis. Foods. 2023. 12(2). (Line 145)

  1. Line 165: Why no information about the column is provided?

Response: Thanks for the reviewer's comment, the corresponding content has been supplemented. (Line 161-162)

8.Line 181: Homo sapiens is also a species and should be in italics. Please verify this in the entire paper.

Response: Thank you for the reviewer's comment. The corresponding content in the article has been revised. (Line 179)

9.Lines 212-215: No comparison of means was performed? Please clarify which tests were used to compare the data.

Response: Thank you for the reviewer's suggestion. The analyzed method of significance of comparisons has been supplemented in the revised manuscript. (Line 212-213)

10.Line 230: Where does the toxicity potential come from?

Response: Thanks for the reviewer’s suggest. The corresponding unclear content in the article has been deleted. (Line 230)

11.Figure 1: Why did the authors not test higher concentrations? An inversion may be happening after the highest concentration, or not.

Response: Thanks for the question. In this experiment, we mainly wanted to understand the semi-inhibitory concentration (IC50) of the peptide against the two enzymes, so we mainly focused on the point where the inhibition rate was 50%. We also did a higher concentration point and found that the inhibition was close to 100% and there was no turning point, which is consistent with the usual inhibition experiment results.

12.Line 264: Explanation to why low-binding energy peptides were chosen has to be added.

Response: Thanks for the comment. The affinity between the ligand and receptor increases as the binding energy decreases. The lower binding energy meaning the better binding capacity. The corresponding content has been added in the revised manuscript. (Line 260-261)

13.Line 372: Where does this comment come from? Which data supports it?

Response: Thanks for the reviewer’s suggest. The corresponding unclear content in the article has been deleted. (Line 372)

14.The fact that prediction suggested that peptides do not present toxicity is interesting, but still toxicity tests are the ultimate check. Why did the authors not check it?

Response: Thanks for the reviewer’s suggestion, it is true that toxicity tests is needed to for the peptides. In the current work, we have mainly carried out preliminary screening, and the use of software is a more recognized preliminary screening method. We will add toxicology experiments to subsequent works. Thank you very much.

15.Authors defend in the discussion that peptides from soy could be used inside their food systems against diabetes but the IC50 was not tested in the original food matrix. So, in future work the effect of these peptides should be checked in the original matrix.

Response: Many thanks to the constructive suggestions of the reviewers, we will complement the inhibition experiments with real samples in the next phase of the experiment.

16.Line 383: First phrase is out of context and does not make a lot of sense. What does nutrition have to do with the paper?

Response: Thanks for the reviewer’s suggest. The corresponding content in the article has been deleted. (Line 382)

Reviewer 2 Report

Comments and Suggestions for Authors

Dear Authors,

yours is certainly an interesting manuscript but it has some problems that need to be resolved.

1) page 3 lines 110-124 This is a misprint please delete it

2) in the materials section please introduce a brief description of peptide solution  preparation

3) the experimental plan is not sufficiently explained. I.e. what are sample groups as in lines 139 and 152? Please provide a more detailed explanation. of sample preparation, numerosity and grouping.

Author Response

RESPONDS TO REVIEWER #2 COMMENTS:

1.Page 3 lines 110-124 This is a misprint please delete it

Response: Thanks for the reviewer’s suggest. The corresponding content in the article has been deleted. (Line 109)

2.In the materials section please introduce a brief description of peptide solution preparation.

Response: Thanks for the reviewer’s constructive suggest. The corresponding content in the article has been revised. (Line 122-126)

3.The experimental plan is not sufficiently explained. I.e. what are sample groups as in lines 139 and 152? Please provide a more detailed explanation. of sample preparation, numerosity and grouping.

Response: Thanks for the reviewer’s comments. The corresponding content has been added to the article. (Line 122-127 and 129-136)

Reviewer 3 Report

Comments and Suggestions for Authors

I consider that this is an interesting paper that fits appropriately to the scope of the journal. However some considerations should be previously taken into account by authors namely:

- Lines 91-92. Authors state that “Soybean has high nutritional value, which is rich in protein and contains 20 essential amino acids”. However, this is not completely true due that its nutritional value is limited by the low content of the sulphur amino acids and specifically of methionine which is the essential one. Thus the nutritional value of the soybean protein is really limited by its low content in methionine.

- Do authors have the idea to study the reported biological activity of these SPs in diabetic animals and finally in diabetic humans? And therefore in the future development of functional foods enriched in them?

- Paragraphs form line 109-124 correspond to information to authors. Please erase them in the revised manuscript.

- Line 137. How many samples of soybean peptides were assayed for performing the study?

- Did authors check if the physical (e.g. heat, cold, pressure, etc.) and/or chemical (e.g. change of pH) treatment of soybean’ proteins as denaturation agents modify the reported biological activity of studied SPs? In affirmative case, do these treatments (e.g. heat, cold, pH, etc.) change the content of the SPs or even the peptide segments?

- Which do authors think that would be the most appropriate food to be enriched with the studied SPs in order to develop a functional food with antidiabetic effect?

- Figure 2. I recommend enlarging its size.

- The references’ list have to be unified and adapted to that reported in instructions for authors in relation to manuscripts submitted to be published in Foods.

                * Sometimes authors use capital letters and other lowercase ones to refer to the         Journal’s name. They use abbreviate or full names…

Author Response

RESPONDS TO REVIEWER #3 COMMENTS:

1.Lines 91-92. Authors state that Soybean has high nutritional value, which is rich in protein and contains 20 essential amino acids. However, this is not completely true due that its nutritional value is limited by the low content of the sulphur amino acids and specifically of methionine which is the essential one. Thus the nutritional value of the soybean protein is really limited by its low content in methionine.

Response: Thanks for the reviewer's comments. The corresponding content in the paper has been modified. (Line 91-92)

2.Do authors have the idea to study the reported biological activity of these SPs in diabetic animals and finally in diabetic humans? And therefore in the future development of functional foods enriched in them?

Response: Thanks to reviewers for comments and replies. Whether the new oligopeptides are biologically active in diabetic animals or patients is the future research direction, and the method of in vitro experiments is a preliminary screening strategy, and we have plans to study the effect of these peptides on blood sugar in vivo through animal experiments, and eventually hope to be able to be applied to functional foods. Thank you very much for your suggestion.

3.Paragraphs form line 109-124 correspond to information to authors. Please erase them in the revised manuscript.

Response: Thanks for the reviewer’s suggest. The corresponding content in the article has been deleted. (Line 109)

4.Line 137. How many samples of soybean peptides were assayed for performing the study?

Response: Thanks for the questions. In this experiment, we first prepared one soybean protein hydrolyzed peptide solution, and then identified 9246 peptides by mass spectrometry from it. And finally, two peptides were selected for enzyme inhibition experiments based on scoring and virtual screening methods.

5.Did authors check if the physical (e.g. heat, cold, pressure, etc.) and/or chemical (e.g. change of pH) treatment of soybean proteins as denaturation agents modify the reported biological activity of studied SPs? In affirmative case, do these treatments (e.g. heat, cold, pH, etc.) change the content of the SPs or even the peptide segments?

Response: Thank you for your valuable advice. It is necessary to discuss whether physical or chemical factors have a significant impact on the composition and activity of soybean peptides. In general, small molecule peptides are considered to have certain stability and biological activity, and have been used in many functional foods or drugs [1,2], and we will further explore the influence of these factors on peptide structure and activity in the next experiments.

  • Sreelekshmi, P.J.; Devika, V.; Aiswarya, L. S.; et al., Recent Advances in Bioactive Peptides as Functional Food for Health Promotions and Medicinal Ap-plications. Protein Pept. Lett. 2023, 30(8), 626-639
  • Fosgerau, K.; Hoffmann, T. Peptide therapeutics: current status and future directions. Drug Discov. Today. 2015, 20(1), 122–128.

6.Which do authors think that would be the most appropriate food to be enriched with the studied SPs in order to develop a functional food with antidiabetic effect?

Response: Thank you for raising the question. We would like to add those peptides as an oral liquid or powder to beverages or liquid foods, before eating large amounts of starchy foods.

7.Figure 2. I recommend enlarging its size.

Response: Thanks for the reviewer's comments. The corresponding content in the paper has been revised. (Figure 2)

8.The references list have to be unified and adapted to that reported in instructions for authors in relation to manuscripts submitted to be published in Foods.

Response: Thanks for the reviewer's comments. The corresponding content of references in the paper has been modified. (References)

Round 2

Reviewer 3 Report

Comments and Suggestions for Authors

Accept.